# Isthmin1 Upregulation in the Intestinal Microenvironment During *Salmonella* Typhimurium Infection: Identification and Characterization of Isthmin1-Producing Cells

**DOI:** 10.3390/microorganisms13061280

**Published:** 2025-05-30

**Authors:** Gustavo Alberto Jaimes-Ortega, Tania Angeles-Floriano, Guadalupe Rivera-Torruco, Julio Cesar Almanza-Pérez, Juan Xicohtencatl-Cortes, Eduardo Hernández-Cuellar, Oscar Medina-Contreras, Ariadnna Cruz-Córdova, Ricardo Valle-Rios

**Affiliations:** 1Immunology and Proteomics Research Laboratory, Children’s Hospital of Mexico “Federico Gómez” (HIMFG), Mexico City 06720, Mexico; gelet321@gmail.com (G.A.J.-O.); lupitariveratorruco@gmail.com (G.R.-T.); 2Experimental Biology Postgraduate Program, Department of Biological and Health Sciences, Metropolitan Autonomous University (UAM), Mexico City 09310, Mexico; 3University Research Unit UNAM-HIMFG, Research Division, School of Medicine, National Autonomous University of Mexico, Mexico City 04510, Mexico; tania.angeles@yahoo.com.mx; 4Subdirección de Diagnóstico Clínico y Departamento de Laboratorio Clínico, Hospital Infantil de México Federico Gómez, Mexico City 06720, Mexico; 5Departamento de Fisiología, Biofísica y Neurociencias, Centro de Investigación y de Estudios Avanzados (CINVESTAV), Mexico City 07360, Mexico; 6Cell Therapy Core, Vitalant Research Institute, San Francisco, CA 94102, USA; 7Department of Laboratory Medicine, Medical Center, University of California, San Francisco, CA 94102, USA; 8Departamento de Ciencias de la Salud, Universidad Autónoma Metropolitana Unidad Iztapalapa, Av. San Rafael Atlixco 186, Col. Leyes de Reforma 1A Sección, Alcaldía Iztapalapa, Ciudad de México 09310, Mexico; jcap@xanum.uam.mx; 9Laboratorio de Investigación en Bacteriología Intestinal, Unidad de Investigación en Enfermedades Infecciosas, Hospital Infantil de México Federico Gómez, Ciudad de México 06720, Mexico; juanxico@yahoo.com; 10Laboratorio de Biología Celular y Tisular, Departamento de Morfología, Universidad Autónoma de Aguascalientes, Aguascalientes 20100, Mexico; edgar.hernandez@edu.uaa.mx; 11Epidemiology, Endocrinology & Nutrition Research Unit, Children’s Hospital of Mexico “Federico Gomez”, Mexico City 06720, Mexico; omedina@himfg.edu.mx; 12Laboratorio de Investigación en Inmunoquímica, Hospital Infantil de México Federico Gómez, Mexico City 06720, Mexico; ariadnnacruz@yahoo.com.mx

**Keywords:** Isthmin1, small intestine, hematopoietic stem cells, *Salmonella* Typhimurium

## Abstract

Isthmin1 (ISM1) is a constitutively secreted cytokine produced by barrier tissues and different immune cell types. Importantly, ISM1 is also expressed by cells with a hematopoietic stem cell phenotype in the lung and plays a role during hematopoiesis. Under inflammatory conditions, ISM1 levels are also altered. Given that the intestine is one of the tissues with the highest ISM1 gene expression, in this work, we characterized the immunophenotype of ISM1-producing cells in the small intestine of uninfected or *Salmonella* Typhimurium-infected mice. We found a variety of ISM1+-expressing cells, including CD45+ cells, EpCAM+ cells, and, importantly, we also found different subsets of cells carrying hematopoietic stem cell markers (LSKs) expressing ISM1, and their frequency was perturbed during infection. Finally, we also found that isthmin1 is secreted into the intestinal lumen, and its concentration was increased during *S.* Typhimurium infection. Our findings suggest that increased luminal ISM1 secretion during mucosal infection may serve as a potential novel biomarker of pathogen-mediated inflammation.

## 1. Introduction

Barrier tissues, such as the lung and intestinal epithelium, constitute the body’s first line of defense against the external environment. These dynamic structures not only provide physical barriers but also play a crucial role in regulating the passage of nutrients, ions, and other substances, while preventing pathogen and toxin entry [1]. The integrity and optimal function of barrier tissues are essential for maintaining homeostasis and preventing the development of various diseases, from respiratory infections and inflammatory bowel diseases to allergies and cancer [2].

The immune response in barrier tissues is a complex and finely regulated process involving a variety of cells and molecules. Under conditions of homeostasis, resident immune cells, such as alveolar macrophages in the lung and immune cells in the lamina propria of the intestine, exert constant vigilance to detect and eliminate any potential threats without triggering excessive inflammation [3]. This surveillance is based on the interaction between immune cells and epithelial cells, as well as on the production of cytokines and other molecules that maintain the immune balance [2].

However, when a pathogen or toxin manages to penetrate the barrier, the immune response is activated more aggressively. Immune cells mobilize to the site of infection, releasing proinflammatory cytokines and other molecules that promote elimination of the pathogen and repair of damaged tissue [2]. This inflammatory response is essential to fight infection, but, if not properly regulated, it can cause tissue damage and contribute to the pathogenesis of various diseases [4,5].

In recent years, the crucial role of extramedullary hematopoietic stem cells (HSCs) in the mucosal immune response has been recognized. Although extramedullary HSCs have been shown to contribute to hematopoiesis under normal conditions, their function during infectious processes and their interaction with the intestinal microbiota are not yet fully understood [6,7]. It has been proposed that extramedullary HSCs may play a role in modulating the immune response by generating immune effector cells and regulating cytokine production [8,9].

Our research group has identified a secretion protein, Isthmin1 (ISM1), which is expressed in cells with extramedullary HSC phenotype in the lung [10,11]. We have observed that ISM1 levels decrease during infections and that these cells express pattern recognition receptors (PRRs), suggesting that they may play a role in modulating the immune response [11]. In addition, ISM1 has been described to interact with GRP78, a receptor that plays a crucial role in cellular homeostasis and signaling [12].

These data suggest that ISM1 could function in regulating the hematopoietic process as has been described in zebrafish, but also its role could be related to protective features, similar to the function of antimicrobial proteins. Another recently described function of ISM1 is its immunoregulatory function. It has been reported that ISM1 is an endogenous anti-inflammatory protein expressed in high amounts in mouse lungs and bronchial epithelial cells; there, it plays a fundamental role in the control of inflammation, leading to apoptosis of alveolar macrophages [12]. These proposed functions of ISM1 may be the result of its interaction with GRP78, its highest-affinity receptor [12,13]. Furthermore, it seems that ISM1 limits the inflammatory outcome during intratracheal LPS instillation since KO *Ism1* mice showed increased immune cells arrival to the lung and increased pulmonary permeability [12].

Some secreted proteins in the intestine combine antimicrobial and immunoregulatory functions and are essential for the body’s overall defense. Defensins, for example, are antimicrobial peptides that not only destroy pathogens by forming pores in their membranes but also modulate the immune response by attracting immune cells and regulating cytokine production. Cathelicidins, another group of peptides, exhibit broad antimicrobial activity, including the ability to induce apoptosis in infected cells, and simultaneously regulate inflammation and immune response through cytokine modulation and interaction with immune system cells. Lysozyme, an antimicrobial enzyme that degrades bacterial cell walls, also plays a role in immunomodulation by influencing inflammation and antigen presentation. Acute phase proteins, such as C-reactive protein, facilitate the opsonization of pathogens and act as markers of inflammation, modulating the immune response and the activity of the complement system. Finally, interferons, especially type I interferons, possess antiviral properties that inhibit viral replication and, at the same time, regulate the adaptive immune response by influencing T and B cell activation and cytokine production [14,15].

While there are efforts to underscore the function of this novel cytokine, studies about ISM1 in the gut are missing. In this work, by using a battery of antibodies, we identified that ISM1 is produced by different subsets of differentiated CD45+ cells and by epithelial cells. Importantly, we also found a group of cells with a hematopoietic stem cell phenotype that were perturbed in a model of *Salmonella* Typhimurium-infected mice. Finally, ISM1 luminal levels were slightly increased, suggesting a role for ISM1 during bacterial infection.

## 2. Materials and Methods

### 2.1. Animals and Ethics Statement

Male C57BL/6 mice (5–8 weeks old) were obtained from Envigo RMS S.A. All animal experiments were performed according to protocols approved by Mexican NOM-062-ZOO-1999 (SAGARPA) and in agreement with the Guide for the Care and Use of Laboratory Animals of the National Institutes of Health (NIH) and internal guidelines. The protocol was approved by the Mexican Children’s Hospital Federico Gomez ethics committee. This study was carried out with the approval and under the guidelines of the Institutional Review Boards (IRBs) of the Ethics Committee of the Mexican Children’s Hospital Federico Gómez.

### 2.2. Isolation of Intestinal Lamina Propria and Epithelial Cells

Mice were euthanized by cervical dislocation. To obtain the cells of interest, the stomach was separated from the upper small intestine by cutting the pyloric sphincter. The mesentery was removed mechanically, and the ileocecal valve was cut to release the entire small intestine from the large intestine. Peyer’s patches were removed along the surface of the small intestine, and the intestine was opened longitudinally. A portion of the fecal content was collected and stored at −70 °C until use. Mucus was removed by washing with PBS at 37 °C. The small intestine was cut into approximately 1.5 cm pieces and placed in Falcon tubes with 30 mL of Hanks (In vitro, San Luis Potosí, San Luis Potosí, México)/5% FBS (In vitro) and 0.5 mM EDTA (Sigma-Aldrich, MilliporeSigma, S.t. Louis, MO, USA. The mixture was shaken at 250 rpm for 20 min at 37 °C in a Bioshaker(model SHKE6000, Thermo Fisher Scientific, Asheville, NC, USA). The liquid was removed by filtration through a 100 μm mesh, and the 1.5 cm pieces were recovered. These pieces were resuspended in 30 mL of DMEM (Gibco, Thermo Fisher Scientific, Grand Island, NY, USA)/FBS with 2 mM EDTA and digested with 20 mg of Collagenase I (Sigma-Aldrich, MilliporeSigma, S.t. Louis, MO, USA) and 10 mg of DNase I (Sigma-Aldrich, MilliporeSigma, S.t. Louis, MO, USA) for 20 min at 37 °C with constant shaking. Subsequently, the cell suspension was filtered through a 70 μm mesh, and cells were collected by centrifugation at 1500 rpm for 5 min at 4 °C. To enrich the proportion of epithelial cells, an initial wash was performed using Hanks solution without EDTA, following the previously described protocol. After filtration, the tissue was homogenized by dissection with scissors and resuspended in 1 mL of PBS supplemented with EDTA. Following a 10 min incubation, the tissue digestion protocol was continued with 20 mg of Collagenase I and 10 mg of DNase I for 20 min at 37 °C with constant shaking, and the previously described protocol was followed.

### 2.3. Flow Cytometry

Flow cytometry was performed by using the following monoclonal antibodies: AF700 anti-CD45, PercPCy5.5 anti-CD19, PercPCy5.5 anti-TER119, PercPCy5.5 anti-CD8a, PercPCy5.5 anti-NK1.1, PercPCy5.5 anti-FceR1a, PercPCy5.5 anti-CD34, BV510 anti-EpCAM, APC anti-Sca-1, APC-Fire anti-c-kit, anti-NK1.1, PE anti-ISM1, PE anti-IgG2b, PercePCy5.5 anti-CD3e, PercPCy5.5 anti-Ly6G, PercPCy5.5 anti-CD11b, FITC anti-Sca1, APC/Fire anti-c-kit, PE/Dazzle anti-CD34, BV605 anti-CD150, and BV510 anti-EpCAM, all from Biolegend (San Diego, CA, USA). Live cells were detected using ViaKrome 808 Fixable Viability Dye from Beckman Coulter. Data acquisition was performed on a CytoFLEX LX Flow Cytometer (Beckman Coulter, Inc., Brea, CA, USA) and analyzed using CytExpert v 2.5.0.77.

Flow cytometry images were acquired by using the BD FACSDiscover S8 Cell Sorter (Becton, Dickinson and Company, Franklin Lakes, NJ, USA) with BD CellView Image Technology and BD SpectralFX.

### 2.4. S. Typhymurium Infection

A chronic infection with *Salmonella* Typhymurium was performed as already reported [16]. A *S.* Typhymurium colony was grown in LB liquid medium, and the following day, two groups of mice were inoculated orogastrically with 1 × 10^7^ bacteria. The feces were recovered to determine the correct infection, and the CFU content was analyzed by seeding on a plate with BBL CHROM (BD). The mice were kept under observation.

### 2.5. Protein Extraction from Feces

Three fecal samples were extracted by adding 5 mL extraction buffer (0.1 M Tris, 0.015 M NaCl, 1.0 M Urea, 1.0 mM CaCI, 0.1 M Citric Acid Monohydrate, 5 mg/mL BSA, and 0.25% Gentamycin Sulfate at pH 8.0) into 100 mg of sample. Samples were vortexed for 10 min and filtered through a 0.2 um filter. Samples were assayed using the Isthmin-1 ELISA Kit LEGEND MAX (BioLegend; San Diego, CA, USA).

### 2.6. ISM1 Protein Measurement

Levels of human ISM1 in the soluble fraction were determined using the Isthmin-1 ELISA Kit LEGEND MAX (BioLegend; San Diego, CA, USA). Six samples were collected in buffer and centrifuged for 10 min at 1000× *g* within 30 min of collection. The ELISA plate was read at 450 nm using a plate reader within 30 min of stopping the reaction. The color intensity is directly proportional to the ISM1 levels in the samples. The concentration of ISM1 in the samples was extrapolated from the ISM1 standard curve.

### 2.7. Statistics

Mouse data are reported as mean ± SEM and were analyzed by unpaired Student’s *t*-test. GraphPad PRISM version 8.0, GraphPad Software Inc., (La Jolla, CA, USA) was used for all these analyses.

## 3. Results

### 3.1. ISM1 Is Expressed in Different Cell Subsets in the Small Intestine

To establish the lymphogate region (Figure 1a), singlets were selected using complexity parameters (SSC-A on the *Y*-axis and SSC-H on the *X*-axis) and size parameters (FSC-A on the *Y*-axis and FSC-H on the *X*-axis). Viable cells were selected using the negative region of the Viakrome dye (Figure 1a, bottom). To better characterize ISM1+ cells, we performed flow cytometry analysis on small intestine cells, and we found around 0.7% of ISM1-producing cells. Due to the high variability in small intestine cells, we decided to measure ISM1 expression in CD45+, CD45+Lin+ cells, the latter included antibodies coupled to PercPCy5.5 against CD19, TER119, CD8a, NK1.1, CD3e, Ly6G, and CD11b (Lin+ cells); interestingly we found that most of ISM1+ cells were CD45+Lin+ compared to CD45+Lin- cells (Figure 1b,e). Interestingly, we found that a high frequency of myeloid cells expressing Ly6G also expressed ISM1 (Figure 1c). Furthermore, we also found that epithelial Epcam+ cells expressed ISM1, which was the least representative ISM1-expressing population (Figure 1e). Finally, we were able to detect ISM1+ cells by flow microscopy. In Figure 1f, CD45+ and EpCAM+ cells expressing ISM1 are depicted together with the intracellular marker Lamp1.

### 3.2. Identification and Characterization of ISM1+ Hematopoietic Stem-like Cells in the Small Intestine

We previously observed a subset of hematopoietic stem-like cells expressing ISM1 in the lung [11], and it was recently observed that the human small intestine harbors a subset of CD34+ hematopoietic stem cells [9]. Therefore, we decided to investigate whether hematopoietic stem-like cells could express ISM1. First, we searched for ISM1 expression in CD45+Lin- cells together with the individual expression of CD34, c-kit (CD117), Sca-1, and CD150, and we found a small fraction of CD45+Lin- cells expressing CD34, c-kit, or Sca-1; we also detected a very small fraction of CD45+Lin-CD150I+SM1+ cells (Figure 2a), suggesting that some ISM1+ cells may have a immunophenotype of hematopoietic stem cells. Furthermore, we looked for the presence of Sca-1 and c-kit expression to obtain a fraction of cells we called ISM1+LSK (CD45+Lin- ISM1+ c-kit+ Sca-1+); importantly the majority of the ISM1+LSK+ cells also expressed CD34 (Figure 2b), indicating that, in the small intestine, there is a fraction of ISM1 hematopoietic stem-like cells. Finally, we confirmed the co-expression of CD45, Sca-1, and ISM1 by ImageStream in single cells (Figure 2c).

### 3.3. Modulation of ISM1 Cell Frequency During Salmonella Typhimurium Infection in the Small Intestine

Since ISM1 expression changed during inflammatory processes involving chronic diseases but also during pathogenic challenges [9,17], we decided to investigate whether ISM1 expression in the small intestine is perturbed during infection. Thus, we orogastrically infected C57BL/6 with *S.* Thyphimurium during 48 h (Figure 3a). Mice were sacrificed after 48 post-infection, and intraluminal contents and cells were collected. We found that the percentage of total intestinal cells expressing ISM1 increased significantly compared to uninfected mice (Figure 3b). Therefore, we analyzed different ISM1+ cell subsets. We found that the fraction of CD45+ISM1+ cells increased significantly during infection (Figure 3b) and, importantly, ISM1+LSK cells were also significantly increased (Figure 3c), suggesting that the fraction of hematopoietic stem-like cells expressing ISM1 was perturbed during infection. We also examined the impact of infection on subpopulations of cells with a hematopoietic stem-like phenotype that expressed ISM1 by analyzing different progenitor cell markers. The results showed that the CD45+Lin-CD34+ISM1+ population tended to decrease during infection (Figure 3d), while the CD45+Lin-CD150+ISM1+ populations significantly increased during infection (Figure 3e). The CD45+Lin-CD117+ISM1+ and CD45+Lin-Sca-1+ISM1+ populations did not show statistically significant changes (Figure 3f,g). These findings suggest that *Salmonella* infection specifically affects the CD45+Lin-CD150+ISM1+ cell population, and likely the CD45+Lin-CD34+ISM1+ population as well, which could have implications for hematopoiesis during the immune response.

### 3.4. Increased ISM1 Secretion in the Intestinal Lumen During Salmonella Typhimurium Infection

Since our data suggested a role for ISM1 during inflammatory responses, and because ISM1 is a cytokine involved during inflammation, we wondered whether the increased frequency of ISM1+ cells observed during *S.* Typhimurium infection may correlate with an increase in ISM1 secretion. We quantified ISM1 levels in the intraluminal contents of the small and large intestines of infected and uninfected mice. We found that control mice showed basal levels of ISM1 in the small intestine. However, ISM1 levels were lower in the large intestine, suggesting that ISM1 cells may be located at the lamina propria of the small intestine. Importantly, the infected mice presented a significant increase in the levels of ISM1 (Figure 4), suggesting a possible role for ISM1 in the response to infection.

## 4. Discussion

For the first time, here, we reported the presence of the novel cytokine Isthmin1 (ISM1) in intestinal cells, expanding our understanding of the distribution and function of this protein. This protein is emerging as a highly relevant molecule in multiple biological processes, from embryonic development to metabolic regulation and tumor response. Its dynamic expression in diverse organisms and tissues suggests a coordinating role in normal growth and development. ISM1 not only influences energy metabolism, regulating glucose, lipid, and protein synthesis, but also modulates cellular processes and fundamental functions, such as apoptosis and angiogenesis, making it a key player in tumor progression. Its ability to influence the immune response and its association with various congenital malformations underscore the complexity and importance of this protein in the biology of organisms [18].

The co-expression of ISM1 with characteristic markers of hematopoietic stem cells (HSCs) in the intestine suggests a crucial role in the regulation of intestinal hematopoiesis. Although more detailed studies are required, preliminary findings indicating the expression of Toll-like receptors (TLRs) in cells with this phenotype in the lung support the hypothesis that these cells can respond to microbial stimuli [11].

Our data showed an increase in the LSK ISM1+ and CD45+Lin-150+ISM1+ cell populations, indicating that the infection may also affect the homeostasis of intestinal ISM1+ HSCs. Given the importance of intestinal HSCs in maintaining immune homeostasis and tissue regeneration, it is tempting to speculate that ISM1 could act as a local factor influencing the proliferation, differentiation, and survival of these cells. To fully elucidate the role of ISM1 in intestinal hematopoiesis, future studies should focus on analyzing the differential expression of ISM1 in different HSC subtypes, as well as evaluating the impact of its modulation on the proliferation, differentiation, and function of these cells in in vitro and in vivo models.

Since ISM1 expression was significantly upregulated during the LPS challenge [17], we decided to measure ISM1 levels. Interestingly, ISM1 levels were also significantly elevated in the small intestine infected with *Salmonella*, suggesting a potential role for ISM1 in response to pathogenic challenges. Given that ISM1 is produced by both immune cells and epithelial cells, it is likely that ISM1 functions not only as a barrier or defense cytokine but also as a regulator of inflammation. Importantly, our findings are consistent with previous work by Ngan Nguyen, who reported increased ISM1 levels in the bronchioalveolar lumen following intratracheal LPS administration. These observations support the idea that ISM1 expression may be upregulated during inflammatory processes to prevent hyperinflammation.

On the other hand, the absence of ISM1 has been associated with the development of chronic inflammatory diseases in the lung, underscoring its importance in maintaining tissue homeostasis [19]. ISM1 may act as a soluble stress sensor that regulates HSC proliferation and differentiation in response to inflammatory stimuli.

The idea that ISM1 plays multiple roles in the gut is not surprising, as many proteins, such as mucins and defensins, possess remarkable functional versatility. This multifunctionality is common in proteins found in complex environments, like the gut, where they must respond to a wide range of stimuli. This multifunctionality is an evolutionary adaptation that allows proteins to respond flexibly to the complex demands of the gut microenvironment. Post-translational modifications, such as phosphorylation and ubiquitination, as well as protein–protein interactions, confer ISM1 and other gut proteins the ability to modulate their activities in response to different stimuli. Moreover, recent studies have shown that ISM1 is significantly overexpressed in colorectal cancer (CRC) tissues and in the blood of patients, correlating with poor prognosis [19]. Therefore, the proper study of Isthmin-1 in intestinal diseases, such as inflammatory bowel disease, Crohn’s disease, and various types of intestinal cancer, could open the possibility of using this molecule as both a diagnostic biomarker and a potential therapeutic tool. Although a deeper understanding of the mechanisms by which ISM1 interferes with various signaling pathways is still needed, its involvement in these processes suggests a promising avenue that warrants further exploration. Understanding the multifunctionality of ISM1 opens new perspectives for the development of targeted therapies for gut diseases.

## 5. Conclusions

Isthmin1 is expressed in the small intestine of mice under homeostatic conditions. The cells that produce it are diverse. ISM1 is expressed by cells with a hematopoietic stem phenotype, including c-kit, CD34, Sca-1, and CD150 expression, which represent the so-called LSK cells. Differentiated cells also express ISM1, such as EpCAM+, CD45+, and Ly6G+ cells. Furthermore, under bacterial infection, populations with a hematopoietic stem phenotype are disrupted, such as ISM1+ LSK and CD45+ Lin-150+ ISM1+ cells, which were increased during infection. On the other hand, ISM1+ levels in the intraluminal contents of the small intestine were also increased during *S.* Typhimurium infection. This study establishes a possible role of ISM1 in intestinal hematopoiesis and the immune response. Our results suggest that ISM1 is a multifunctional molecule with a key role in homeostasis and host defense. However, further research is needed to fully understand the molecular mechanisms underlying its function and therapeutic potential.

## Figures and Tables

**Figure 1 microorganisms-13-01280-f001:**
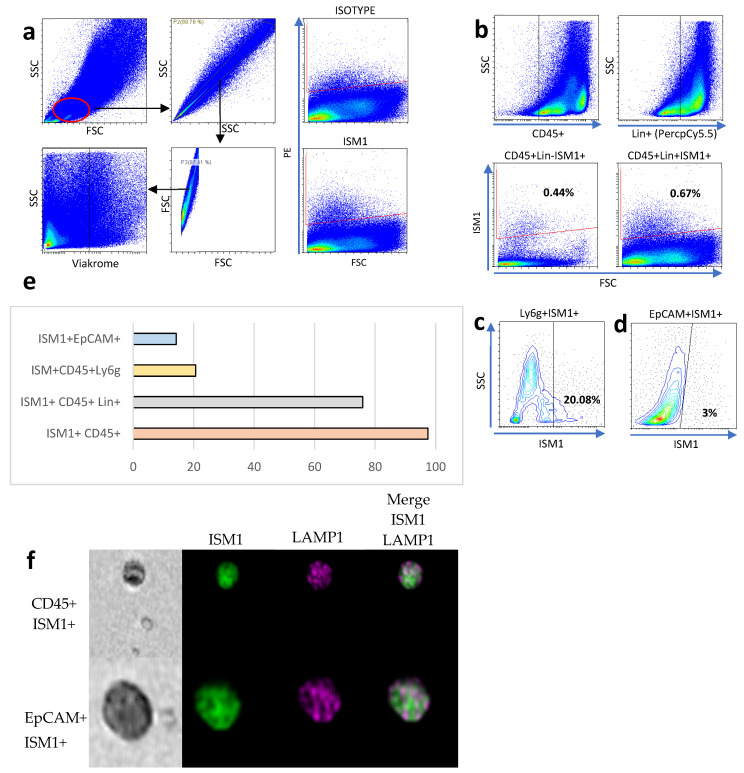
ISM1 expression in small intestine cell subsets. (**a**) Flow cytometry analysis strategy for identifying ISM1+ cells in the small intestine. (**b**) Percentage of ISM1+ cells in CD45+Lin- and CD45+Lin+ populations. (**c**) Co-expression of ISM1+ and Ly6G cells in myeloid cells. (**d**) Percentage of ISM1+ cells in the EpCAM+ population. (**e**) Summary of ISM1 distribution in the different cell subsets analyzed. (**f**) Representative flow microscopy images showing ISM1 co-expression with LAMP1 in CD45+ and EpCAM+ cells. A total of 10,000 events were collected, from which 2000 CD45-positive images and 840 EpCAM-positive images were selected and analyzed. Data represent mean ± standard deviation from at least three independent experiments.

**Figure 2 microorganisms-13-01280-f002:**
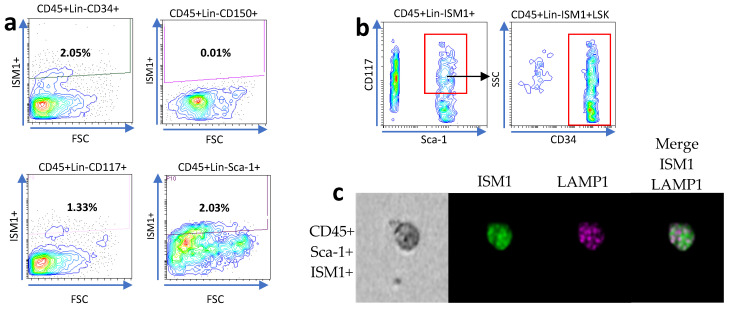
Characterization of ISM1+ hematopoietic stem-like cells in the small intestine. (**a**) Flow cytometry analysis showing the expression of CD34, c-kit (CD117), Sca-1, and CD150 in CD45+Lin- ISM1+ cells. (**b**) Co-expression of CD34 with the ISM1+LSK fraction (CD45+Lin- ISM1+ c-kit+ Sca-1+). (**c**) Flow microscopy (ImageStream) images showing the co-expression of CD45, Sca-1, and ISM1 in individual cells. From a total of 10,000 collected events, 1600 Sca-1–positive images were identified and analyzed. The data represent three independent experiments.

**Figure 3 microorganisms-13-01280-f003:**
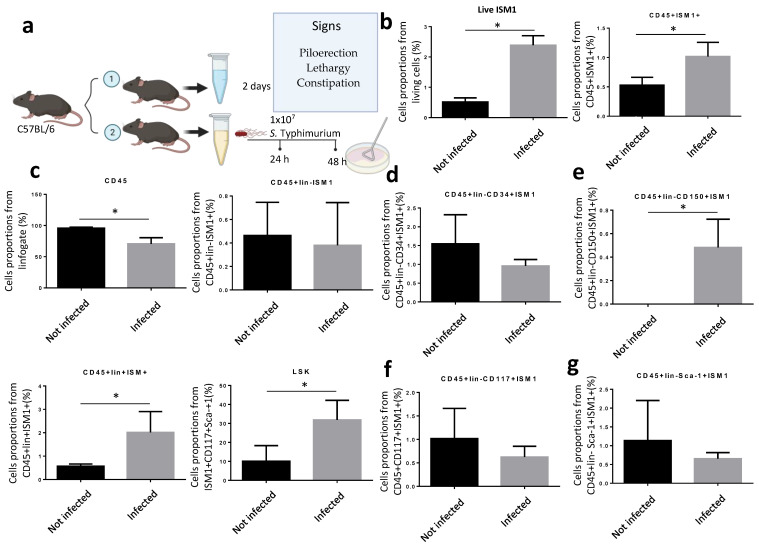
Modulation of ISM1 expression during *S.* Typhimurium infection. (**a**) Schematic of the experimental design for orogastric infection with *S.* Typhimurium in C57BL/6 mice. (**b**) Percentage of total intestinal cells and ISM1+CD45+ cells in uninfected and 48 h infected mice. (**c**) Percentage of ISM1+LSK cells in uninfected and 48 h infected mice. (**d**) Percentage of CD45+Lin-CD34+ISM1+ cells in uninfected and 48 h infected mice. (**e**) Percentage of CD45+Lin-CD150+ISM1+ cells in uninfected and 48 h infected mice. (**f**) Percentage of CD45+Lin-CD117+ISM1+ cells in uninfected and 48 h infected mice. (**g**) Percentage of CD45+Lin-Sca-1+ISM1+ cells in uninfected and 48 h infected mice. Data represent the mean ± standard deviation from at least three independent experiments. * *p* < 0.05.

**Figure 4 microorganisms-13-01280-f004:**
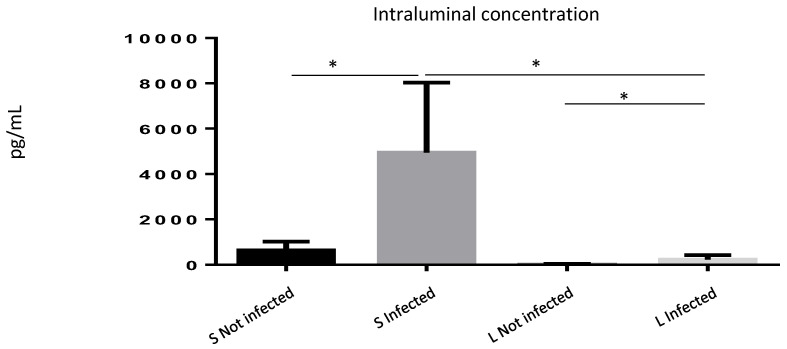
ISM1 levels in the intraluminal contents of the intestine during *Salmonella* Typhimurium infection. Quantification of ISM1 levels in the intraluminal content of the small (S) and large (L) intestines of uninfected and *S.* Typhimurium-infected mice at 48 h post-infection. Data represent the mean ± standard deviation of at least three independent experiments. * *p* < 0.05.

## Data Availability

The original contributions presented in this study are included in the article. Further inquiries can be directed to the corresponding author.

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
