# Peer review of "Isthmin1 Upregulation in the Intestinal Microenvironment During Salmonella Typhimurium Infection: Identification and Characterization of Isthmin1-Producing Cells"

_microorganisms, 2025, doi:10.3390/microorganisms13061280_

Round 1
Reviewer 1 Report
Comments and Suggestions for Authors
Manuscript by Gustavo Alberto Jaimes-Ortega et al. titled "Isthmin 1 upregulation in the intestinal microenvironment during Salmonella typhimurium infection: Identification and characterization of Isthmin 1 producing Cells" is an original study showing increased Isthmin 1 levels in experiments with mice infected by S. Typhimurium. I have no major objections to the manuscript. The main shortcomings mostly concern formatting and are listed below.
Apparently, the authors are not very familiar with the Salmonella nomenclature. The serovar name should be capitalized and not italicized. Please check and correct this throughout the text, including the title. Also, after the first mention, Salmonella should be abbreviated to S. Typhimurium. Additionally, italicize "Salmonella" throughout the text (e.g., lines 43, 168, 170, 258, etc.).
Citations should also be formatted correctly - as numbers in square brackets.
"ISM1" should be spelled out at first mention (line 80), not at the seventh (line 95).
Subheadings in the Materials and Methods section should be numbered as 2.1., 2.2., etc.
In line 171, the number 7 should be in superscript.
Also, specify the manufacturers of reagents in brackets (e.g., BBL CHROM and others).
Additionally, add a section 2.7. on statistical analysis: which methods were used, what software, etc.
Many figures are not referenced in the text: for example, 1a, 1d, and there is no panel "e" in Figure 1. The same applies to Figure 3 - only 3b and 3c are referenced.
In line 224, "Figure 2" is repeated twice.
In Figure 4, the label "a" should be removed since there is only one panel.
Author Response
Please, find a file attached with the answers. Thank you

Reviewer 2 Report
Comments and Suggestions for Authors
This study has characterized the immunophenotype of ISM1-producing cells in the small intestine of uninfected or Salmonella typhimurium-infected mice. it found a variety of ISM1+ expressing cells, including CD45+ cells, EpCAM+ cells. it also found different subsets of cells carrying hematopoietic stem cell markers (LSK) expressing ISM1 and their frequency was perturbed during infection. Finally, it found that isthmin1 is secreted into the intestinal lumen and its concentration was increased during Salmonella infection. The findings suggest that increased luminal ISM1secretion during mucosal infection may serve as a potential a novel biomarker of pathogen-mediated inflammation. The experiment was well designed. The paper was well written. The following revision could improve the quality of the paper.
Major comment
The outcomes suggest that ISM1 is a multifunctional molecule with a key role in homeostasis
and host defense. What is the clinical significance? What we need to do in the future? The authors need to disscuss this in the paper.
Minor comments
L125,135, 156, etc, the section title, such as Animals and Ethics Statement and ect should be bold.
L132-133, please do not written only 2-3 rows as a paragraph, which can be combined with the other content.
All figures, how many replicates has been used for the analysis? Please add these information in the footnote.
The quality of presentation of Figure 3 needs to be improved.
All the P value should be written in italic. Such as L254, etc.
L254, L269, etc. What is the p<0.05 means? The sentence is not completed.
Figure 4 looks not good. Please try to improve the presentation quality of the data.
L349, some of the reference did not followed the standard of the journal. Please check them carefully.
Author Response

(The authors gave the same response as above.)
